# Unraveling the Intertwined Effect of pH on *Helicobacter pylori* Motility and the Microrheology of the Mucin-Based Medium It Swims in

**DOI:** 10.3390/microorganisms11112745

**Published:** 2023-11-10

**Authors:** Clover Su-Arcaro, Wentian Liao, Katarzyna Bieniek, Maira A. Constantino, Savannah M. Decker, Bradley S. Turner, Rama Bansil

**Affiliations:** 1Department of Physics, Boston University, Boston, MA 02215, USA; clover13@mit.edu (C.S.-A.); wtliaonn@gmail.com (W.L.); kasia99987@gmail.com (K.B.); maira.ac@gmail.com (M.A.C.); savannah.m.decker.th@dartmouth.edu (S.M.D.); 2Department of Biological Engineering, Massachusetts Institute of Technology, Cambridge, MA 02139, USA; bsturner@mit.edu

**Keywords:** *Helicobacter pylori*, motility, mucus, gastric mucin, microrheology, bacteria swimming

## Abstract

The gastric pathogen, *Helicobacter pylori* bacteria have to swim across a pH gradient from 2 to 7 in the mucus layer to colonize the gastric epithelium. Previous studies from our group have shown that porcine gastric mucin (PGM) gels at an acidic pH < 4, and *H. pylori* bacteria are unable to swim in the gel, although their flagella rotate. Changing pH impacts both the rheological properties of gastric mucin and also influences the proton (H+)-pumped flagellar motors of *H. pylori* as well as their anti-pH sensing receptors. To unravel these intertwined effects of acidic pH on both the viscoelastic properties of the mucin-based mucus as well as the flagellar motors and chemo-receptors of the bacterium, we compared the motility of *H. pylori* in PGM with that in Brucella broth (BB10) at different pH values using phase contrast microscopy to track the motion of the bacteria. The results show that the distribution of swimming speeds and other characteristics of the bacteria trajectories exhibit pH-dependent differences in both media. The swimming speed exhibits a peak at pH 4 in BB10, and a less pronounced peak at a higher pH of 5 in PGM. At all pH values, the bacteria swam faster and had a longer net displacement in BB10 compared to PGM. While the bacteria were stuck in PGM gels at pH < 4, they swam at these acidic pH values in BB10, although with reduced speed. Decreasing pH leads to a decreased fraction of motile bacteria, with a decreased contribution of the faster swimmers to the distributions of speeds and net displacement of trajectories. The body rotation rate is weakly dependent on pH in BB10, whereas in PGM bacteria that are immobilized in the low pH gel are capable of mechano-sensing and rotate faster. Bacteria can be stuck in the gel in various ways, including the flagella getting entangled in the fibers of the gel or the cell body being stuck to the gel. Our results show that in BB10, swimming is optimized at pH4, reflecting the combined effects of pH sensing by anti-pH tactic receptors and impact on H+ pumping of flagellar motors, while the increase in viscosity of PGM with decreasing pH and gelation below pH 4 lead to further reduction in swimming speed, with optimal swimming at pH 5 and immobilization of bacteria below pH 4.

## 1. Introduction

The human stomach presents one of the harshest environments due to the high acidity of its gastric juice secretion and various aspartate proteases and digestive enzymes, which are crucial for metabolizing food and destroying microbes. While the combination of gastric juice and the gastric mucus barrier is quite effective in sterilizing and protecting the host from bacteria and infections, the gastrointestinal pathogen, *Helicobacter pylori* has adapted to this particularly challenging environment and can move across the mucus layer. *H. pylori* colonizes on the epithelial surface, and even deep in the gastric glands, leading to the development of diseases such as gastric ulcers and gastritis, and is associated with gastric cancer [1]. The pH of the stomach is an important factor in controlling the colonization and pathogenic effects of *H. pylori* [2] and many questions about the effect of pH on *H. pylori* remain poorly understood. In this paper, we focus on how varying pH influences the motility and swimming behavior of *H. pylori* across the viscoelastic gastric mucus. 

The pH of the stomach varies over a 24 h period and also in different locations of the stomach. Measurements of pH in normal human subjects using a catheter with a miniature pH-sensing electrode show that over a 24 h period, the mean intragastric pH measured in the lumen and near the mucosal surface was close to 2, rising after meals for about 45 min to pH values > 3, 4 or 5 [3]. When the electrode was pushed into the mucosa under endoscopic control the pH rose above 4. These observations are consistent with the existence of a pH gradient across the mucus layer due to the co-secretion of bicarbonate and the acidic gastric juice [4]. The pH of the mucus varies from an acidic pH of 2–4 on the luminal side to a neutral pH of 6–7 on the epithelial surface [4,5,6]. Microelectrode measurements of the pH at different depths from the gastric epithelial surface in vivo in rats reveal that when exposed to a luminal pH of 2–3, the pH remains close to the luminal pH value for about 85–100 μm in the mucus layer, and then steadily increases, reaching 6.8–7 at the epithelial surface [5,6]. Ross et al. [5] measured pH with microelectrodes at different distances from the luminal surface of rat stomachs in vivo exposed to HCl solution at pH 2 and observed that the pH remained close to 2 up to a distance of about 160 μm in the mucus gel and then steadily increased reaching a maximal pH close to 6.7 at a distance of about 400 μm from the luminal surface. Thus their observation shows a gradient from pH 2 to 6.7 over a mucus thickness of 240 μm equivalent to a H^+^ concentration gradient of ~0.5 × 10^−4^ moles/L per μm. Schade et al. [6] used pentagastrin for the stimulation of acid secretion in vivo in rats and observed the pH gradient dropping from a pH of 7.2 at the epithelial surface to pH 2 over a distance of about 115 μm from the epithelial surface, corresponding approximately to a concentration gradient of 10^−4^ moles/L per μm. A similar pH gradient was recorded at luminal pH 3, dropping from pH 7.3 at the epithelial surface to pH 3 over a distance of 85 pm from the epithelial cell surface. The pH gradient has been modeled by using an approach based on the Nernst–Planck equation to calculate the ion transport of bicarbonate and H^+^, assuming that the H^+^ is bound to the mucus gel [7]. 

The mucus pH gradient plays a key role in guiding *H. pylori* to navigate away from the acidic luminal surface towards the neutral epithelial surface [8] using chemotaxis mechanisms that involve various Tlp chemoreceptors for bicarbonate, anti-pH tacticity and urea [8,9,10], as well as a cascade of proteins to relay the signal, to the flagellar motors [11], thereby influencing the swimming motility of the bacteria [12,13]. The bacteria attach to the epithelial surface using various adhesins and evoke an immune response that causes the inflammation of the surrounding tissue. 

The variation in pH directly influences the viscoelastic properties of the glycoprotein mucin predominantly consisting of MUC5AC, which is responsible for the viscoelastic barrier properties of mucus. Gastric mucin undergoes a pH-dependent solution-to-gel transition at pH 4 [14], forming a viscoelastic gel below pH 4. Our group has previously reported that in the absence of added urea, *H. pylori* shows no translational motion in purified porcine gastric mucin (PGM) solutions at pH < 4, although the flagella could be observed rotating in bacteria trapped in the mucin gel network [15,16]. The trapping of bacteria in the mucin gel is not surprising given that it has a structure of fiber bundles and pores at many length scales, ranging from around 200 nm, as seen in AFM images of mucus [17], up to the micron range, as seen in SEM images of mucus [16]. As is well known, *H. pylori* employ a urease secretion mechanism to hydrolyze urea, which produces NH_3_ and CO_2_ [1], enabling the bacterium to survive in an acidic environment. In the absence of urea, *H. pylori* survives between a pH of 4.5 and 7.0 in vitro and does not grow at a pH less than 3.5 without urea, i.e., it behaves as a neutrophile in standard buffers, not surviving extreme acidity or alkalinity [18]. In our previous work, we show that the addition of urea also leads to a gel-to-solution transition as the pH increases above 4 [15], enabling the bacteria to swim across the mucus layer. However, it is not clear from this previous study [15] how much of the observed pH-dependent change in motility is due to the gelation of PGM and how much is due to the influence of pH on the flagellar motors.

Previous studies of varying pH on motility in several bacteria such as *B. subtilis*, *E. coli*, and *Salmonella* [19,20,21,22] have shown that motility is not affected much by changing the external pH over the rage of pH 8 to 5. Instead, it depends on the internal, i.e., cytoplasmic, pH of the cell, which can be modified by the addition of weak acids in the medium that are known to permeate biological membranes. For example, in *E. coli* and *Salmonella* in the presence of acetate, decreasing pH causes swimming speeds, and rotation rates measured by tethering cells to decrease over the pH range of 7 to 5 [19], with the motor rotation stopping completely around pH 5. This observation was interpreted as an interference of the increase in intracellular H^+^ concentration with the protons released from the torque-generating units. In contrast, *B. subtilis* swims well at pH 5.5 by maintaining a more or less constant PMF over the range of pH 5–7 [21]. *Salmonella* displayed an anti-pH tactic behavior and enhanced tumbling in the presence of weak acids [19]. A study of the torque-speed relationship showed that while high-speed rotation at low load was impaired in the presence of a weak acid [22] zero-speed torque was not affected. None of these effects were observed in the absence of weak acid. It is not clear whether these observations apply to *H. pylori* as unlike *E. coli* and *B. subtilis*, it has a bundle of multiple unipolar, membrane-sheathed flagella. The *H. pylori* motor has evolved to adapt to a highly viscous and somewhat acidic environment, using urease to regulate its cytoplasmic pH. It is capable of high torque generation, having 18 stators per motor [23,24] as opposed to 11 in *E coli* and *Salmonella* and 8–11 in *B. subtilis*.

Previous studies of *H. pylori* swimming in culture broth BB10, methyl cellulose and PGM [25,26,27], as well as cell rotation rates in BB10 and PGM [28] were all conducted at pH 6. These studies showed that at pH 6 *H. pylori* exhibit a run-reorient-reverse swimming mechanism. Constantino et al. [28] were able to track flagella and cell motion in a few swimming bacteria using higher magnification, and they showed that the cell body of *H. pylori* rotates in the opposite sense to the flagella rotation as it swims. They also noted that the speed is faster when it moves in the forward direction as a puller with flagella rotating counterclockwise and body rotating clockwise as opposed to the reverse motion as a pusher. The reverse swimming behavior of some *H. pylori* mutants has also been investigated at a neutral pH. Antani et al. [13] compared the circular tracks of bacteria swimming close to a glass surface in native *H. pylori* in broth at pH 7 with a mutant lacking the phosphorylated response regulator, CheY-P, which is known to promote the clockwise rotation of flagella in *H. pylori.* They show that *H. pylori* swims at a faster speed when they move in the forward direction with flagella rotating counterclockwise acting as a pusher compared to the reverse run with flagella rotating clockwise acting as a puller. Howitt et al. [12] showed that mutants of *H. pylori* lacking ChePep, a protein that is required for the polar localization of some of the chemotaxis components, show more frequent reversals and sustained swimming in the reverse direction, implying that ChePep plays an important role in controlling the directional persistence of motility. Several studies show *H. pylori* bacteria swimming away from regions of low pH due to the anti-pH tactic response of Tlp protein receptors [8,10,29], although detailed analysis of the swimming trajectories of bacteria was not conducted.

In this paper we present a detailed study of the pH dependence of the translational and rotational motion of *H. pylori* across the entire range of pH values reflected in the mucus barrier to understand how the effect of pH on the H^+^ pumped flagellar motors coupled with the anti pH-tactic response at low pH influences the motility of *H. pylori*. By comparing the motility of *H. pylori* in aqueous Brucella broth (BB10) vs. PGM over the relevant pH range of 7–3 we are able to separate the influence of increased H^+^ concentration on the flagellar motors and pH sensing chemoreceptors of *H. pylori* from that due to the pH dependent viscoelastic properties of mucin. Our results show that due to the competing effects of anti-pH tactic receptors and impairment of flagellar function on lowering pH, the swimming speed displays a non-monotonic dependence with optimal swimming at pH 4 in BB10. The additional effect of the pH-dependent viscosity is evident with reduced swimming speed below pH 5 in PGM as opposed to pH4 in BB10. The bacteria can still swim in BB10 at pH 3, whereas they rotate in a more or less stuck position in the PGM gel, with the rotation rate increasing due to mechanosensing.

## 2. Materials and Methods

### 2.1. H. pylori J99 Culturing Conditions

The J99 WT strain of *H. pylori* was provided by Prof. Sara Linden of the University of Gothenburg, Sweden. For details of the strain and its source see [30,31]. J99 was cultured initially from frozen stocks (stored in −80 °C freezer) on Brucella agar (Brucella Medium Base, Oxoid, Basingstoke, Hampshire, UK) supplemented with 10% sheep blood, 1% IsoVitox (Oxoid), 4 mg/L amphotericin B, 10 mg/L vancomycin and 5 mg/L trimethoprim for 48 h then re-plated for an additional 48 h on Brucella agar. Each culture was then inoculated in liquid media (BB10) containing 10% fetal bovine serum and 90% Brucella broth for 7–10 h on the shaker, then diluted with broth and incubated for additional 12–16 h on the shaker to optimize the number of motile bacteria. All agar or broth cultures were maintained at 37 °C under microaerobic conditions in a tri-gas environment using BD Biosciences GasPak EZ Campy Container System Sachets to increase the levels of CO_2_ as we did not have access to a tri-gas incubator. The manufacturer states that after adding the sachets to the GasPak Container, the levels of oxygen can vary from 6 to 16%, while levels of CO2 can vary from 2 to 10%. (BD Biosciences, San Jose, CA, USA). The concentration of bacteria in the liquid culture was monitored by measuring the absorbance using a spectrophotometer. The liquid culture is added to each sample when the bacteria reach the exponential growth phase *OD_600_* = 0.6–0.7.

### 2.2. PGM Preparation

PGM was collected from pig stomach epithelium and purified with Sepharose CL-2B column chromatography, followed by density gradient ultracentrifugation, described in Celli et al. [14] and more fully in the original references therein. Lyophilized PGM was weighed, and the appropriate amount of PGM was dissolved in sterile water to prepare a final solution at 15 mg/mL after the addition of 10% bacterial or bead sample and 10% buffer. Typically, 40 μL of sterile water was added to 0.75 mg of PGM and mixed using a vortex mixer for 10 to 20 min until there was no visible un-hydrated PGM (white spots). This solution was allowed to hydrate and equilibrate for 48 h at 4 °C and was used for a maximum period of one week. When ready to use, either 5 μL of 0.1 M phosphate–succinate buffer (adjusted to various pH values using HCl) or 5 μL of BB10 with HCl was added to adjust pH and thoroughly mixed with a positive displacement pipette. The buffered PGM solution was incubated at 37 °C for at least 40 min before 5 μL of bacterial or beads sample could be added.

### 2.3. pH Calibration in BB10 and PGM

BB10: For each pH, 100 μL of bacteria liquid culture or particle solution was added to 800 μL of fresh BB10, followed by a gradual addition of HCl and BB10 until a desired pH level and a final total volume of 1 mL were reached. A 3-point calibrated handheld pH meter was used to monitor the pH of each sample.

PGM: For each pH, 2.5 μL of bacteria liquid or particle solution was added to 20 μL of PGM solution in buffer, followed by a gradual addition of buffer, HCl and BB10 until the desired pH level and a final total volume of 25 μL. The pH of each sample was monitored using Hydrion pH strips. We noted that when the pH was over 2–6, the final pH of the PGM solution was higher than the initial buffer pH. When the external buffer was at a pH of 2, 4, 5, 6 and 7, the PGM solution had a pH of 3.7, 4.6, 5.5, 6.1 and 6.7, respectively [32]. This is in keeping with the H^+^ binding properties of mucin, a negatively charged polymer [33,34]. 

The measurements of the viscosity of BB10 and PGM using particle tracking microrheology [30,31,32] are included in Appendix A.

### 2.4. Motility: Phase Contrast Microscopy Live Cell Imaging, Cell Tracking

Sample preparation: Each PGM or BB10 sample was incubated at 37 °C for 45 min. Bacteria were cultured in a liquid broth (BB10) to an *OD_600_* of 0.6–0.7 then added to each sample to produce a 10% bacteria mixture by volume. The bacteria mixture was incubated at 37 °C under microaerobic conditions using GasPak systems and constant agitation for 45 min before measurement. A 10 μL volume of each sample was applied to a glass microscope slide with a secure seal spacer and sealed with a coverslip.

Phase contrast imaging: The samples were imaged immediately at room temperature using an Olympus IX70 inverted phase contrast microscope equipped with a 40× objective lens (0.65 NA), a halogen light source, and an Andor Zyla 5.5 sCMOS camera at 33 fps and 6.5 μm pixel size. Videos of bacteria swimming in mid-plane between the coverslip and microscope glass slide were acquired for 9 s using Micro-Manager 1.4 open source acquisition software, as in the earlier work from our laboratory [30,34]. 

All imaging experiments at different pH in the two media, BB10 and PGM were conducted from the same batch of bacteria to minimize variation from batch to batch. However, this sequential approach limited the number of different pH values that could be examined as bacteria cannot be used for very long. To image body rotation, we used a 100× oil-immersion objective lens (1.25 NA) and an Andor Zyla 5.5 sCMOS camera at 100 or 200 fps as described in [28]. Three-second videos of bacteria swimming in mid-plane between the coverslip and microscope glass slide were acquired.

Cell tracking and analysis of trajectories: Bacteria and particle trajectories were tracked using the PolyParticleTracker MATLAB routine to determine the instantaneous position in 2 dimensions. Using the position vector of each bacteria, we calculated the instantaneous speed as the displacement per unit time between two consecutive frames and the direction in which the bacterium was traveling as the angle φ(*t*) of the 2-dimensional velocity vector. Further analysis was performed by segmenting each trajectory into runs and reorientations using a modified version from Hardcastle [35] of the method developed by Theves et al. [36]. 

Reorientation events were determined by looking for large changes in the direction of the velocity vector that were much greater than the change in direction due to the rotational diffusion of a similar-sized particle. The instantaneous speed (v_ins_) was calculated from the displacement divided by the time between two consecutive time frames, and the direction in which the bacterium was traveling by the change in the angle of the displacement vector. The run speed (v_run_), was defined as the average speed over a linear path between two reorientations or reversal events. Reorientation angles between runs are denoted by θ_re_ and if larger than 140° they were identified as reversals. The reversal frequency was calculated as the number of reversal events in a trajectory divided by the time duration of a trajectory and the percent of reversals from the number of reorientations with angle changes larger than 140° relative to the total number of all reorientations. 

### 2.5. Body Rotation and Cell Shape Analysis

The movie of each bacterium was individually and manually cropped using ImageJ. The cell body contour and the centerline of each bacterium were extracted and aligned using CellTool Python software [37]. For each trajectory, we can obtain numerous images of the cell shape of an individual bacterium and we select the one with the largest axial length (most in-plane image) as explained in Constantino et al. [28] to obtain the cell shape parameters. The body rotation rate of each bacterium was measured by monitoring the change in the alignment angle of the bacterium and the time between two maximum points, as described in Constantino et al. [28].

All numerical analyses were performed using Matlab.

## 3. Results

### 3.1. pH Dependence of Swimming Speed Distributions of H. pylori in BB10 and PGM

To determine how the pH and the viscoelasticity of the environment influence the motility of *H. pylori*, we tracked bacteria from the *H. pylori* J99 strain swimming in BB10 and 15 mg/mL PGM with pH ranging from 2 to 6.3. We performed a detailed analysis of the movies to obtain the bacteria trajectories and instantaneous positions of the bacteria following the methods described by Martinez et al. [27]. Figure 1 shows images of all recorded trajectories from entire movies at different pH values in BB10 and in PGM. We note that the color coding of trajectories is arbitrary; it is not identified with the time where the trajectory began or which bacterium followed which trajectory. Different trajectories could arise from the same bacterium going in and out of the image plane, although trajectories which cross each other were clearly from bacteria moving at different times. In the images of Figure 1, we are displaying only the trajectories from swimming bacteria; those that were not motile were separately counted. From these images, some differences between BB10 and PGM are obvious. First, comparing BB10 and PGM, we observe a larger fraction of linear trajectories in BB10 compared to PGM. Trajectories with turns and reversals can also be seen. The trajectories in PGM appear more helical than in BB10. Secondly, regarding the effect of pH, we note that there are fewer trajectories at pH < 5 in BB10 and pH < 4 in PGM. We found that in BB10 the bacteria swam over the entire range of pH 3–6.3, with a decline in the percentage of motile trajectories with decreasing pH, although some bacteria became immotile and coccoidal at pH 3. In contrast to this, in PGM the bacteria swam only at pH 4 and higher; there were very few swimmers (~5 in total) in the pH 3.5 sample, and this data was not analyzed. 

At pH 3, the bacteria were stuck and observed to rotate without changing their position, i.e., they were not swimming. The percent of motile bacteria was counted by looking at randomly selected frames in the movies. By this method, we estimate that in BB10, there are about 40% motile bacteria at pH 3–4 and around 60% at pH 5–6, whereas in PGM, there were only 12% motile bacteria at pH 4, and around 20–25% motile bacteria at pH 4.5, 5 and 6. These data are reported in Table 1. The reference bacteria from the sample in BB10 were also examined at the same time as the PGM measurements were being conducted and these remained motile confirming that bacteria were viable and that the immobility was due to gelation of PGM and not due to loss of motility in the bacteria.

### 3.2. H. pylori Display Differences in Duration and Length of Trajectories

To characterize the qualitative differences in the trajectories for bacteria swimming in BB10 and PGM at different pH values, we used two parameters to describe each trajectory: the net displacement (d) over the time interval (τ) between the first and last frame. Appendix A shows a contour plot of the probability p (d, τ), where the contours represent curves of equal probability. 

Figure 2 shows the histogram of the distribution of d for BB10 (Figure 2A) and PGM (Figure 2B) at all pH values. The distribution of d extends to larger values for BB10 than PGM, and the average <d> is ~ 2 times larger in BB10 as compared to PGM (Figure 2C,D). In both media, decreasing pH diminishes the occurrence of larger net displacement, with <d> dropping by a factor of ~ 0.4 in BB10 between pH 6 and 5, and a more pronounced drop in PGM by a factor of ~ 0.7 between pH 4.5 and 4. In PGM, no trajectories extend beyond 20 μm at a pH of 4, reflecting the gelation of PGM close to pH 4. The standard deviation *σ*, which is a measure of the breadth of the distribution of d decreases at pH < 5 in BB10, and pH < 4 in PGM. 

Figure 3 shows the distribution of the time duration of the trajectory τ in BB10 (Figure 3A) and PGM (Figure 3B). This distribution is very flat for PGM (Figure 3B) compared to BB10 (Figure 3A) and shows that the bacteria swim for much longer along the trajectory in PGM at pH 4, i.e., they swim very slowly at pH 4 in PGM. 

### 3.3. H. pylori Displays Broad Speed Distributions in PGM and BB10 

The trajectories in Figure 1 show that the bacteria exhibit linear runs of various duration, punctuated by reorientation and reversals, in agreement with our previous reports [27]. To quantify this, the trajectories were analyzed by segmenting the trajectory into runs and reorientations as described in the Materials and Methods section. The distributions of both v_ins_ and v_run_ were obtained from the analysis of a large number of trajectories, typically greater than 200, and in some cases, up to 600 trajectories were analyzed. Figure 4A–D shows the probability distributions of v_ins_ and v_run_ in BB10 and PGM at different pH values. We calculated the average instantaneous and run speeds, <v_ins_> and <v_run_>, as well as the median speeds and the standard deviation *σ* for the entire distribution. The results are summarized in Table 1. As discussed in our earlier work, *σ* is a measure of the width of the distribution and reflects both the temporal variation in speed, as well as the variation due to the polydispersity in the number of flagella, and helical shape and size of the bacteria [27]. 

The speed distributions in Figure 4 are broad and appear to be bimodal in both BB10 and PGM. We refer to these peaks as “slow swimmers” and “fast swimmers”, respectively. The peak position of for the faster swimmers shows a non-monotonic behavior increasing from about 25 μm/s to 50 μm/s as pH decreases from 6.3 to 4 and then decreasing to ~40 μm/s at pH 3 for both v_ins_ and v_run_. The distribution for v_run_ in PGM is considerably narrower than that in BB10, the faster peak is less prominent, and the positions of the peaks are lower than in BB10. Although the peak position for v_run_ in PGM shows a similar trend with varying pH as in BB10, it only increases slightly from ~20 μm/s to 25 μm/s over the pH range from 6 to 4.5 and then drops back down to ~20 μm/s at pH 4. The faster swimmers are most likely the bacteria with a larger number of flagella, as reported in our previous work [27] by comparing mutants with on average one more and one less flagellum [38]; although shape variation between individual cells also contributes to variation in speed. 

The probability distribution of turn angles, θ_re_ at different pH is shown in (Figure 4E,F) for BB10 and PGM, respectively. It shows that the bacteria tend to reorient more frequently and reverse much less in PGM as compared to BB10 at all pH values. The cumulative probability of reversals, estimated from turns between 140° and 180°, is around 0.3 in BB10 but almost an order of magnitude smaller in PGM, ranging from about 0.03 in the solution phase (pH > 4) to 0.07 in the gel phase at pH 3 (Figure 4F). Most of the reorientations in BB10 and PGM are in the range of 25–50° in both PGM and BB10, except at pH 4 in BB10, which shows a different behavior than other pH values in BB10, with a high probability of large angle reorientations. 

Figure 4C,D show the pH dependence of <v_run_> for BB10 and PGM, respectively. The <v_ins_> also show the same trend as <v_run_>; these data are provided in Table 1 but not displayed in Figure 4. The variation in the overall average is influenced by the two-peak character of the distributions. In BB10, <v_run_> displays a clear maximum at pH 4. The variation of <v_run_> with pH in PGM is much less pronounced, and occurs at higher pH of 5.

### 3.4. The Effect of pH on Cell Body Rotation in BB10 and PGM

While there have been some limited, previous results about the effect of pH on swimming speeds, to the best of our knowledge, there is no previous study about the effect of pH on *H. pylori* rotation. To measure the effect of pH on cell body rotation, we imaged the motion of bacteria at high magnification (100×) and fast frame rate (100–200 fps), as was previously performed for *H. pylori* at neutral pH by Constantino et al. [28]. A few frames from the Appendix A provided in the Appendix A show a bacterium rotating while it is swimming in BB10 at pH 4 (Figure 5A). The overall behavior is similar to that reported by Constantino et al. [28] for LSH 100 strain of *H. pylori* swimming at pH 6 in BB10 and PGM. As discussed in [28], the axis of the flagella and that of the cell body are not colinear, in other words, the motion is like a precession. In the movies, we can sometimes see flagella, which enables us to identify the pusher or puller (forward or reverse) motion. A typical contour and alignment axis of the bacterium obtained by CellTool is shown in Figure 5B. Successive contours along the trajectory (Figure 5C) show the bacterium rotating while swimming forward as a pusher (A to B in the figure), then reversing to swim backward (B to C) and reversing again at C to swim in the forward direction to the point D. The closer spacing of successive contours in the segment BC indicates that the bacterium is moving slower while it is in the reverse puller mode as compared to the forward swimming pusher mode in segments AB and CD, which is in agreement with our previous work [28] and the observation of Antani et al. [13] by examining Che-Y mutants swimming in circular tracks close to the glass surface, as described in the Introduction. The alignment angle as a function of time (Figure 5D) shows clear oscillations as well as larger reorientations due to the trajectory changing direction between runs. From these data, we can obtain the speed and frequency over each oscillation as well as the average speed V over the entire trajectory and the average rotation rate Ω. When the bacterium image is fully in the image plane we can obtain good estimates of its length, thickness and helical pitch, and number of turns in the helix. We note that the speed measurement at this high magnification is based on shorter trajectories than the ones from 40× imaging because the depth of focus is smaller and bacteria remain in focus for only short times. Moreover, some fast bacteria move out of focus so rapidly that they cannot be recorded. Due to these experimental limitations, only a few bacteria could be tracked at high magnification, and thus, the swimming speed observed may not be statistically representative of the whole population of bacteria. 

Figure 5E,F shows the cell body rotation rate (Ω) averaged over each trajectory at different pH values in BB10 and PGM, respectively. The rotation rates that we observe for J99 in pH 6 in BB10 and PGM are similar to those reported by Constantino et al. [28] for the LSH100 strain which has the same average number of flagella as J99 [27,30,39]. A comparison of Figure 5E,F shows that the rotation rate in BB10 is more or less unchanged over pH values varying from pH 6 to 4 and then decreases as the pH decreases to 3. The rotation rate in PGM exhibits a more complex behavior as below pH 4, the bacteria do not translate but still rotate [15]. In the pH range where bacteria can swim in PGM, the rotation rate first decreases as pH decreases from 5.5 to 4.5 and then increases, reaching a peak value at pH 3.5 and dropping slightly as pH drops to 3. The increase in rotation rate below pH 4 in PGM correlates with stuck bacteria, suggesting that the bacteria exhibit a mechanosensing response to the increase in viscosity near the solution-to-gel transition on decreasing pH below 4. 

### 3.5. Different Types of Stuck Bacteria 

Figure 6 shows some images of bacteria stuck in different ways, obtained from the Appendix A provided in Appendix A. Along with a few images at different times as indicated (Figure 6A–D), we also show the trajectory of the bacteria by plotting their positions over the entire movie as a single image (Figure 6E–H). Several different modes of sticking are observed as compared to the *H. pylori* bacterium reported in [15] whose cell body was totally stuck so it could not translate, although its flagella rotated at varying speeds. The bacterium in Figure 6A at pH 4 in PGM appears to be stuck at the end of the flagella, and the flagella and the cell body rotate about an axis going through this point. The images of Figure 6B show a pair of bacteria in PGM at pH 4 with one of them swimming in a small circle, while the other is in an orthogonal position. A bacterium at a slightly higher pH of 4.5 in PGM shows a small amount of random translation and reorientation while it rotates (Figure 6C). In contrast, the bacterium in BB10 at pH 4 swims in a straight run (Figure 6D). Thus, it appears that bacteria can become stuck by having either their flagella, their body or both become entangled with the polymeric gel-like medium, or exhibit passive, hindered diffusive motion due to confinement in the pore of a gel.

## 4. Discussion

The pH dependence of the motility of *H. pylori* is governed by three factors, namely (i) the effect of pH on the proto-pumped flagellar motors, (ii) the chemosensing receptors that move it away from acidic pH, and (iii) the viscoelastic properties of gastric mucus that are controlled by gastric mucin. The pH-dependent variation in BB10 reflects only the response to pH of flagellar motors and pH-sensing receptors as the viscosity is independent of pH. On the other hand, in PGM, the effect of pH on flagellar motors and pH-sensing receptors is intertwined with the increase in the viscosity of the gastric mucin with decreasing pH and a solution to gel transition below pH 4. To interpret the effect of the increasing viscosity of PGM on the swimming of *H. pylori*, we plot in Figure 7 the viscosity of BB10 and PGM as a function of pH along with the average run speeds (from Figure 5G,H) and the average body rotation rate (from Figure 6E,F). More details about the viscosity data obtained from particle-tracking microrheology are provided in the Appendix A and refs. [14,27,30,31,32]). We note that for bacteria motility, it is the microviscosity on length scales comparable to the bacteria that is relevant and not the bulk viscosity, which is much higher. The viscosity of PGM at 15 mg/mL increases by almost a factor of 3 as pH decreases from 6 to 4, while that of BB10 remains constant close to the value of water of 1 cP independent of pH. 

As pH decreases from neutral to lower values, the average speed <v_run_> in BB10 increases, reaching a clear maximum around pH 4. The reorientation frequency shown in Figure 4, also shows a markedly different behavior in BB10 at pH 4, with fewer reversals and an increased probability of large angle reorientations. The optimal swimming speed and fewer reversals at pH 4 reflects the combined effect of pH on flagellar motors and anti-pH tactic receptors. It implies that while the anti pH-tactic response would have led to increased motility with decreasing pH [28], the H^+^-pumped flagellar motors become less efficient below pH 4, causing the speed to decrease and large reorientations to increase. The maximum in speed in PGM occurs at a higher value of pH 5 and is much less pronounced as compared to BB10, reflecting the additional effect of increasing viscosity of PGM with decreasing pH which further impedes the swimming speed. The reversal frequency in PGM is much less than in BB10. 

The ratio of the run speeds in PGM relative to BB10 can be used as an indicator of the decrease in speed due to changing viscosity in PGM. The ratio of average run speeds in PGM versus BB10 is ~0.5 at pH 6 and decreases further to ~0.3 at an acidic pH of 4 due to the increase in viscosity of PGM as pH drops from 6 to 4. The fraction of motile *H. pylori* bacteria also declines with decreasing pH, although *H. pylori* can swim in BB10 at lower pH values than other bacteria, having adapted to transit through an acidic environment. The formation of a gel below pH 4 has a dramatic effect on motility, causing the bacteria to be immobilized in the gel at pH 3, as observed previously [15]. 

We found that the body rotation rate is weakly dependent on pH in BB10 from pH 6 to 4 and then increases slightly at pH 3, whereas in PGM, it shows a peak at pH 3.5 correlated to viscosity increase as the pH decreases (see Figure 7B). The decrease in the rotation rate at pH 3 in both PGM and BB10 reflects the impairment of flagella motors. At this low pH, many bacteria were coccoidal in both BB10 and PGM indicating that under such extreme acidity, bacteria are directly impacted by acid. In previous work from our group we have also observed stuck but rotating *H. pylori* bacteria in gelatin gels, but not in solutions of gelatin or polymethylcellulose [27,35]. In this study we observe that bacteria can be stuck in many different ways. For example, they can be physically stuck in the gel network; at a single point (as shown in Figure 6); the entire cell can be stuck while the flagella rotate, as reported in Celli et al. [15]; or trapped in liquid pores in the gel, showing hindered diffusional motion. For stuck bacteria, the interaction with the medium could lead to additional forces and torques that hold the bacteria in place, and this might explain the increase in the rotational frequency of the cell and the loss of translational motion. We also observed that stuck bacteria rotate faster than motile ones, perhaps indicating a mechano-sensing ability of *H. pylori* trapped in a gel network. Mechanosensing has been observed in recent studies of the flagellar motor with varying load and viscosity of the medium [40,41] and they suggest that the bacterial flagellar motor senses the external load and mediates the strength of stator binding to the rest of the motor.

The pH dependence of rotation rates reported here has implications for torque generation and the stator function of *H. pylori*. The forces and torques on the bacterium are proportional to its local speed and angular rotation rate, with the proportionality determined by the viscosity and hydrodynamic drag coefficients which depend on the shape and size of the bacterium. As shown in Figure 6E,F the rotation rate from individual bacteria at any given pH show a large variation (close to a factor of 2.5 for the swimming bacteria in BB10 and PGM), which implies that there is a large variation in the torque exerted at fixed pH due to the variation in the size and number of flagella among individual bacteria. Furthermore the torques will vary with pH proportionately to the rotation rate and the viscosity η, as the hydrodynamic drag coefficients are all proportional to the viscosity. As the observed rotation rates are comparable in BB10 and PGM, we expect that the ratio of the torque of cell rotation T_c_/η in BB10 is comparable to that in PGM. However, the actual torque in PGM will be higher, scaling in proportion to the viscosity, which increases as pH decreases. If we use the actual viscosity of purified PGM then T_c_ would be about two orders of magnitude higher in PGM as compared to BB10. A more realistic estimate is obtained if we take into consideration that the effective viscosity of mucin is reduced due to the active motion of the bacterium [42] and the shear thinning properties of mucin [14]. We have reported elsewhere that the effective viscosity measured from particle tracking in PGM with active J99 *H. pylori* bacteria is reduced by about a factor of 5–10 [30]. In this case, torque T_c_ would only be about 10–20 times higher in PGM than in BB10 at pH 6 and about 20–40 times higher in PGM at pH 4.

The flagella motor of *H. pylori*, like other *ε*-proteobacters, such as *C. jejuni*, has adapted to high torque generation by having a large scaffold of proteinaceous, periplasmic rings to increase the radius of the contact lever point to accommodate the larger number of stators (17 for *C. jejuni* [24,43] and 18 for *H. pylori* [23]). We can estimate the maximum torque for *H. pylori* using the same approach as Beeby et al. [43] by assuming that the torques of all stators are additive and that each stator complex exerts a force of 7.3 pN. For the lever arm, we use 27 nm as the distance between the outer lobe of the C ring and the axis of rotation to obtain a maximum torque of 18 × 7.3 pN × 27 nm ~ 3550 pN. nm per flagellum. As mentioned earlier, the unipolar bundle of the *H. pylori* J99 strain used here has 1–6 flagella with on average 3 flagella [39], implying that *H. pylori* could exert a torque of 1–6 times the maximal torque per flagellum, and that it can vary its motor torque by two orders of magnitude by varying the number of active stators. Such a mechanism is consistent with the effect on stator recruitment observed in a recent study of a *Salmonella* MotA (M206I) mutant, where the number of functional stators is restored by lowering the external pH [22]. Furthermore, periplasmic pH, which is close to external pH, may directly impact motor function as the scaffolding rings are in the periplasmic space. The faster swimmers are more likely to have a larger number of flagella [27] implying that more stator units can be turned on. With decreasing external pH, the proteins in the scaffold structure in the periplasmic space could be impacted and affect the stators.

As mentioned in the Introduction, *E. coli* and *Salmonella* also show the impairment of flagellar function at low pH. The swimming speed of *E. coli* in culture broth showed a similar behavior as *H. pylori*, changing from 20 μm/s to 25 μm/s when the external pH varied from 5 to 8 and exhibited a maximum at pH 6.5, with the motors having stopped rotating when the cytoplasmic pH dropped below 5 [20]. On the other hand, we observe maximum speed for *H. pylori* in BB10 at a lower pH of 4, and the flagella continue to rotate even at external pH 3. The values of the pH at which impairment occurs appear to be lower for *H. pylori* compared to *Salmonella* and *E. coli*, although these results on *E. coli* and *Salmonella* cannot be directly compared to our findings as they varied the internal cytoplasmic pH by adding weak membrane permeable acids [19,20,21]. Differences may also be related to the structural differences in the flagella as *H. pylori* has membrane-sheathed flagella, while the others do not. Moreover, *H. pylori* has adapted to colonize in acidic gastric mucus as opposed to *E. coli* or *Salmonella*, which transit through the stomach but do not colonize there.

## 5. Conclusions

In conclusion, our findings show that changing pH alters the motility of *H. pylori* in both culture broth and gastric mucin. The competing effects of pH on anti-pH tactic receptors and flagellar motors lead to an optimum swimming in slightly acidic conditions. The rheology changes in the viscoelastic PGM gel further impede the swimming speed with the bacteria getting stuck in the gel and being unable to swim freely even though they rotate faster as they mechanosense their gel environment. 

*H. pylori* is an interesting candidate for detailed studies of the pH dependence of torque–speed relationships as the properties of both the medium (viscosity, gelation, specific adhesin binding and polyelectrolytic interactions) and the flagellar motors and chemoreceptors of the bacterium depend on pH. Further investigation to directly measure microrheology in the immediate vicinity of the bacterium would be useful, as the bacterium influences the properties of the medium by urease-mediated hydrolysis of urea to elevate pH locally. Examining the pH dependence of motility in mutants lacking various Tlp and flagellar proteins, or with fluorescently labeled flagellin or other flagellar proteins to directly visualize flagella motion, will address many questions about its swimming mechanism. Measurements of flagellar torque and effects of pH on the stators and scaffolding complex should reveal the dependence of motility on cytoplasmic pH and the transmembrane proton gradient.

The work presented here may have broader relevance to the regulation of acid in the treatment of *H. pylori* infection and colonization of gut bacteria. However, it is important to note that the work reported here was carried out in the absence of urea. *H. pylori* utilizes the urease-mediated hydrolysis of urea to produce NH_3_, which increases the pH. This not only helps the bacteria survive in an acidic environment [18] but also reduces mucin viscosity and de-gels mucin, enabling the bacteria to swim [15]. Preliminary measurements of chemotactic motility under acid and urea gradients are reported in [31] and provide a more realistic model of relevance to gastric physiology. Our results could further stimulate the development and testing of theoretical and computational fluid dynamics models to understand the motion of bacteria in gels and other confined geometries [44], as well as guide the design of artificial biomimetic swimmers using pH to control motility.

Similar studies in other *Helicobacter* spp. would also be relevant to the motility and colonization of these pathogens in humans and other mammals [45] and further advance our understanding of the mechanism by which Helicobacters can breach the gastric mucus barrier and colonize the gastric epithelium causing diseases, such as gastritis, gastric ulcers and even leading to gastric cancer.

## Figures and Tables

**Figure 1 microorganisms-11-02745-f001:**
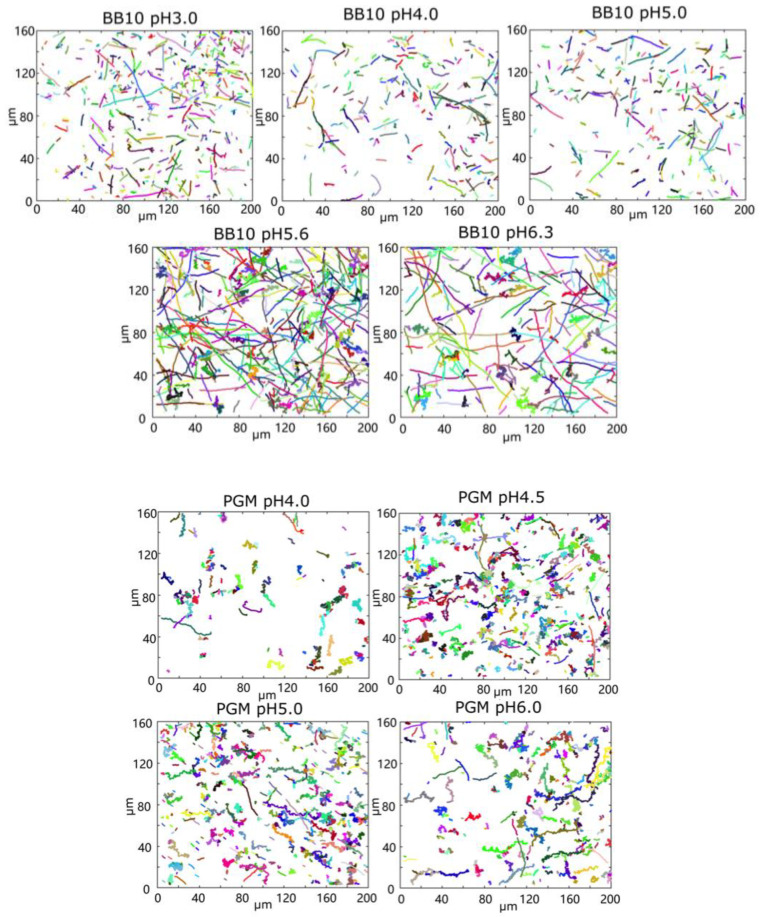
Trajectories of *H. pylori* in BB10 and PGM at different pH values as indicated. The trajectory images were created by stacking frames from a video of 9 s duration recorded at 33 fps to make a composite image.

**Figure 2 microorganisms-11-02745-f002:**
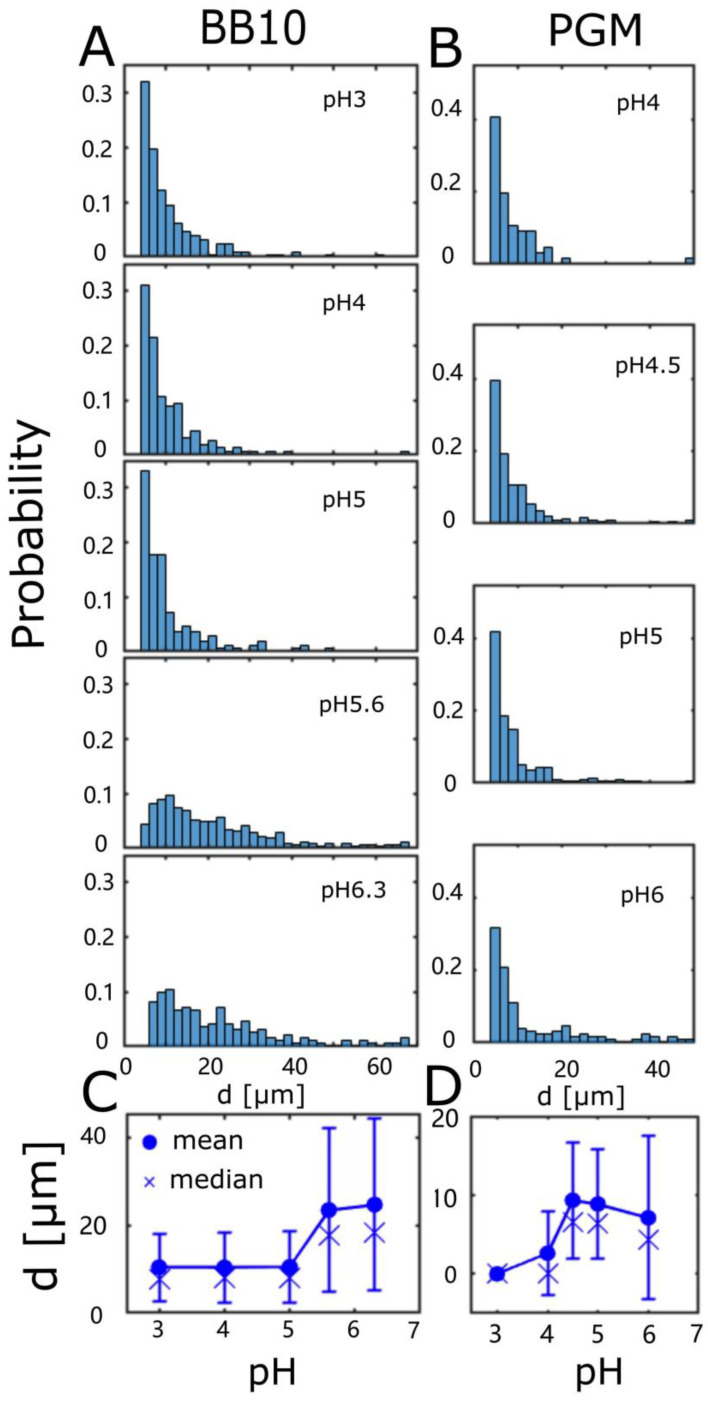
The distribution of trajectory displacements in BB10 (**A**) and PGM (**B**). Histogram of net displacement, d, along the trajectory in BB10 (**A**) and PGM (**B**) at different pH values as indicated, and the dependence of the average displacement (mean and median) on pH in BB10 (**C**) and PGM (**D**). Note that trajectories with d < 4 μm are not displayed as these correspond to bacteria that only display passive diffusional motion.

**Figure 3 microorganisms-11-02745-f003:**
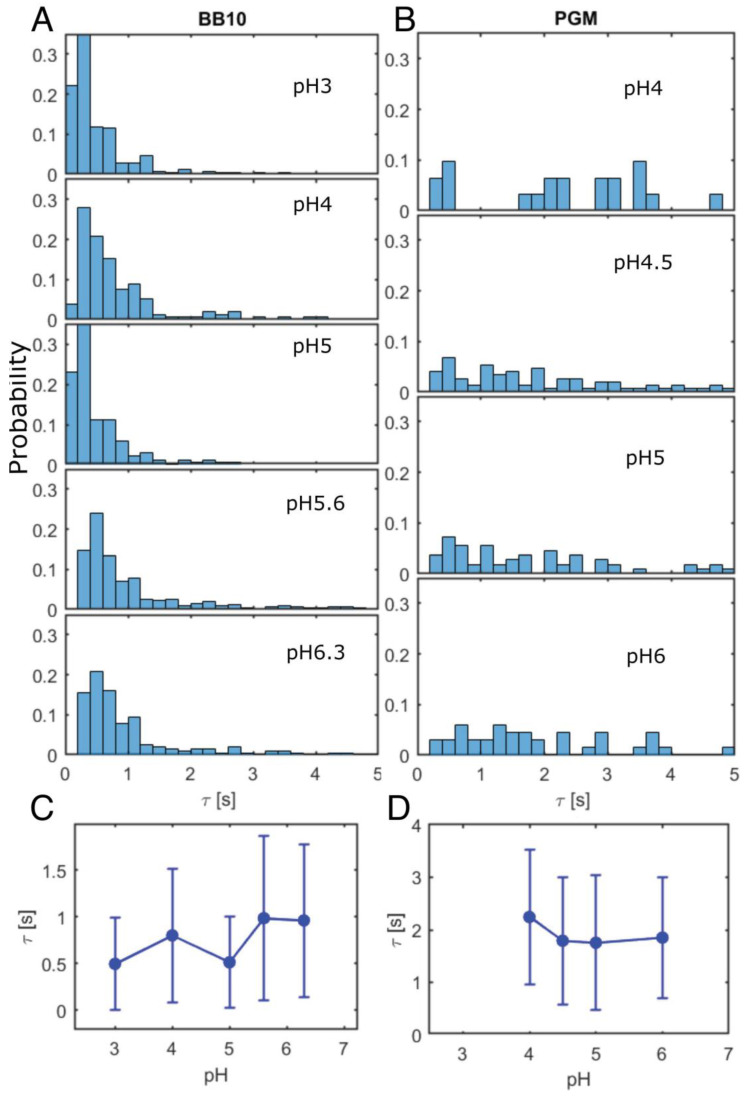
The distribution of trajectory time duration in BB10 (**A**) and PGM (**B**). Histogram of net time duration (τ ) of each trajectory in BB10 (**A**) and PGM (**B**) at different pH’s as indicated. The pH-dependence of the average time duration along with standard deviation *σ* in BB10 (**C**) and PGM (**D**).

**Figure 4 microorganisms-11-02745-f004:**
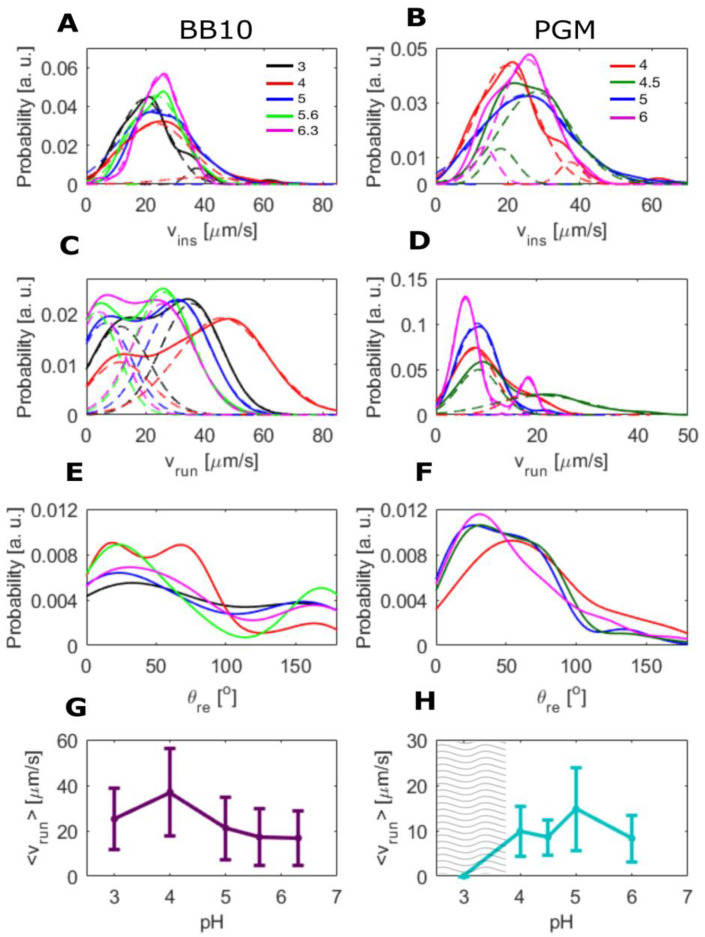
Distributions of swimming speeds and turn angles of *H. pylori* in BB10 and PGM at various pH values. Distribution of the instantaneous speed (v_ins_) in BB10 (**A**) and PGM (**B**) are shown at different pH values as indicated. Distribution of the run speed (v_run_) in BB10 (**C**) and PGM (**D**) are shown at different pH values as indicated. The dashed lines show a two-peak fit to the distribution. The turn angle (θ_re_) distributions in BB10 (**E**) and PGM (**F**) at different pH values as indicated. The average run speeds <v_run_> and standard deviation (*σ*) as a function of pH in BB10 and PGM are shown in (**G**,**H**). The hatched region in **H** represents the gel regime.

**Figure 5 microorganisms-11-02745-f005:**
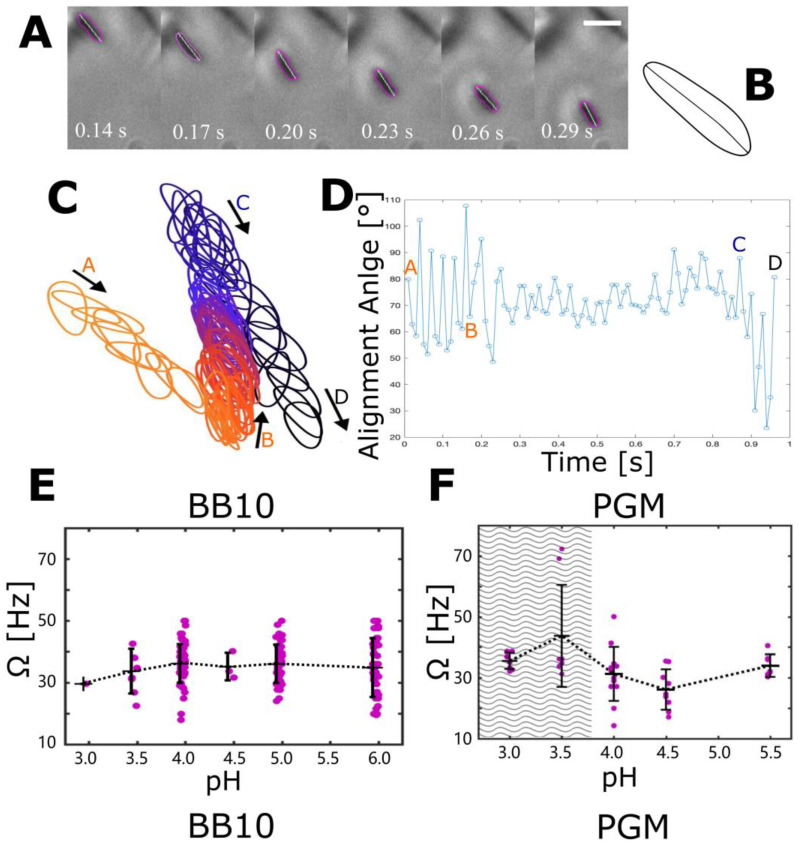
Rotation of cell body while *H. pylori* bacteria swim. (**A**) A series of images from different frames of the video showing a single bacterium at 100× as it rotates while translating in BB10 at pH 4. The contour of the bacterium is highlighted in purple, and its axis is shown in white. (**B**) A typical contour, and the center line was obtained from CellTool. (**C**) Successive contours along the bacterium’s trajectory. A, B, C, and D indicate reorientation events as described in the text. The frames shown in (**A**) are from the motion segment, A to B. (**D**) Alignment angle (relative to an arbitrary direction) showing oscillations as the cell rotates. The movie for this data is provided in the Appendix A. (**E**,**F**) The rotation rate of the cell body (Ω) in BB10 (**E**), and PGM (**F**) at different pH values. The dashed lines are a guide to the eye. The gray-shaded region represents the pH range over which PGM gels and bacteria did not translate but only rotated.

**Figure 6 microorganisms-11-02745-f006:**
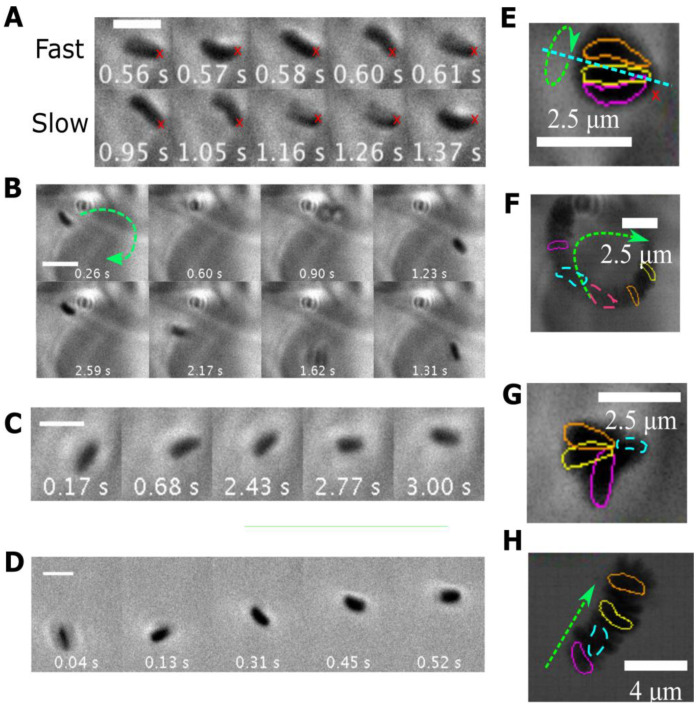
Phase contrast microscopic time-lapse montages and images demonstrating trajectories of various modes of *H. pylori* bacteria stuck in PGM at low pH (**A**–**C**) and a swimming bacterium in BB10 at pH 4 (**D**). (**A**) A stuck bacterium in PGM at pH 4 appears to rotate about a fixed point (indicated by the red x) with fast and slow body rotation rates over time. (**B**) A bacterium is stuck on a fixed circular trajectory in PGM at pH 4 perhaps reflecting proximity to a region of high PGM concentration (dark region). (**C**) A stuck bacterium at pH 4.5 in PGM showing rotations along with small, random translational displacements. (**D**) A swimming bacterium in BB10 at pH 4 shows a clear translational movement while rotating the cell body. The scale bars in (**A**–**D**) indicate a length of 2.5 μm. The time from the video is shown on the images. The dark region in (**E**–**H**) represents overlays of different frames of the movie onto a single image to show the trajectory corresponding to (**A**,**B**,**C**,**D**), respectively. The contours of the bacterium are highlighted in different colors for a few frames. The colors are arbitrarily chosen with the green arrows indicating the direction of motion in time. In image (**G**) the bacteria are colored pink, yellow, blue and orange in order of increasing time, and a green arrow for time is not provided as the motion is random. The movies for this data are provided in the Appendix A.

**Figure 7 microorganisms-11-02745-f007:**
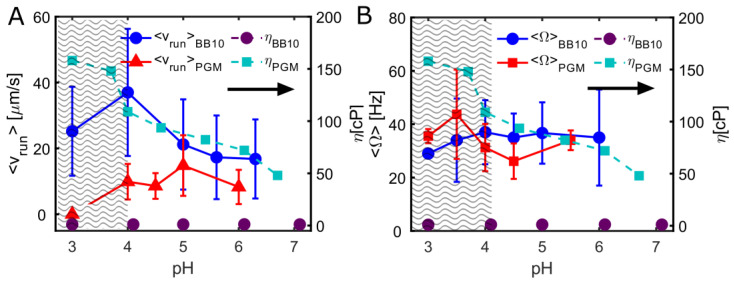
A summary of the effect of pH on *H. pylori* motility. (**A**) The average run speed <v_run_> in BB10 and PGM at different pH values. (**B**) The average rotation rate Ω and PGM at different pH values. In both panels (**A**,**B**), the viscosity η of BB10 and PGM is shown, with the axis on the right, as indicated by the right arrow. The shaded region represents the range of pH over which PGM forms a viscoelastic gel.

**Table 1 microorganisms-11-02745-t001:** Summary of the average, standard deviation σ, and median values of instantaneous and run speeds. The percentage of motile bacteria determined from all the trajectories where bacteria moved and percent reversals are also given.

Medium	pH	<v_ins_>	σ	Median	<v_run_>	σ	Median	Motile	Reversal
		μm/s	μm/s	μm/s	μm/s	μm/s	μm/s	%	%
BB10	3	33	10	34.1	25.2	13.5	27.6	40.4	56.0
	4	47.5	17.1	48.1	37	19.3	39.6	42.3	24.6
	5	29.8	9.3	30.9	21.2	13.7	21.7	60.9	61.5
	5.6 *	25.2	6.5	25.4	17.3	12.7	23.6	61.2	70.7
	6.3 *	24.9	7	25.3	16.8	12	16.8	59.7	56.3
PGM	4	21.9	10.2	20.8	9.9	5.4	8.9	12.4	37.9
	4.5	26.1	11.9	25.6	8.6	3.9	8.5	24.8	13.1
	5	27.1	10.1	26.3	14.8	9.2	10.8	27.3	8.1
	6	23.7	10.9	21.1	8.3	5.2	6.4	21	42.9

* Indicates sample made with pH buffer instead of HCl.

## Data Availability

Data reported here can be obtained by contacting the corresponding author, R.B., at rb@bu.edu.

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
