# Peer review of "Unraveling the Intertwined Effect of pH on Helicobacter pylori Motility and the Microrheology of the Mucin-Based Medium It Swims in"

_microorganisms, 2023, doi:10.3390/microorganisms11112745_

Round 1
Reviewer 1 Report
Comments and Suggestions for Authors
I have tried to read this manuscript carefully in order to discover what is novel about what is being presented. Unfortunately, I am at a loss to say what that would be.
It was already known that PGM forms a gel at pH values below 4. It is well known that all bacteria have trouble swimming through gels when the mesh becomes tight enough. I gather that the pH dependence of swimming for H. pylori is known from previously published work. The pH-dependent tactic behavior of H. pylori has also been described. I therefore fail to see what new information is gained by combining effects of pH on swimming and gelation of PGM in one study. The effects are totally predictable.
This sense of much ado about very little is heightened by the sheer length and detail of the study. All sorts of statistical analyses are employed to describe different aspects of motility, but I do not see that they add much to simple statements of results.
The monopolar flagellar bundle of H. pylori works in either a pusher mode (counterclockwsie [CCW] flagellar rotation) or puller mode (clockwise [CW] flagellar rotation). The speed of swimming is higher in the pusher mode than in the puller mode (Antani et al, ELife, January 2021). Do the changes in swimming speed measured in the current study reflect changes in the speed of flagellar rotation, a change in the proportion of pusher and puller cells, or a combination of both?
To answer that question, it would be useful to look at the behavior of a cheY mutant of H. pylori, which is locked in CCW flagellar rotation, and thus the pusher mode, as a function of pH both in broth and in PGM at different pH values. It would also be interesting to know if puller cells are more apt to get stuck than pusher cells as the level of gelation increases at lower pH. Are cells that are able to penetrate further through the gel ones that are more biased toward the pusher mode, which is what happens in response to an attractant stimulus? There may also be mutants of H. pylori that are locked in CW rotation, and if there are, it would be good to include them in the study as well.
In its current form, the manuscript is far too long and contains a great deal of data analyzed by different statistical methods that add little or nothing to the conclusions. My suggestions for improvement are several fold. 1) Add a study of the behavior of a CCW-locked cheY mutant, and of a CW-locked mutant if it is available, in broth and in PGM as a function of pH. 2) State very clearly what new information has been gained from the study. Just saying untangling the intertwined effects of pH on motility and viscosity is not sufficient; both of those have been reported separately in previous work. 3) Drastically decrease the verbiage and the redundant studies.
What the reader wants to learn is how H. pylori penetrates the mucus layer of the stomach. In its present form, the study contributes little to understanding that process.
Comments on the Quality of English LanguageThe manuscript is twice as long and has twice as many figures as it needs. The English is all right, but the writing is very verbose. A far more concise version would be much more readable. Just one comments on style. I think pH values is more standard and acceptable than pHs.
Reviewer 2 Report
Comments and Suggestions for Authors
The manuscript "Unraveling the intertwined effect of pH on Helicobacter pylori motility and the microrheology of the mucin-based medium it swims in." investigates the effects of pH on Helicobacter pylori motility in two different media - Brucella broth and porcine gastric mucin. The authors have systematically examined how pH impacts motility characteristics such as swimming speed, body rotation rate, trajectory analysis, and fraction of motile bacteria.
The authors have done a commendable job of decoupling the effects of pH on the bacterium versus the medium. The manuscript is clearly written, and the figures are appropriate. I only have a few suggestions to further improve the paper:
· In the introduction, provide some quantitative estimates for the pH gradients reported across the gastric mucus layer (pH 2-4 luminal side, pH 6-7 epithelial surface).
· For the viscosity measurements, it would be helpful to show viscosity vs pH plots for both Brucella broth and gastric mucin in the main text or supplementary information.
· Discuss in more detail in the discussion how your findings compare to previous observations of pH effects on motility for other bacteria like E. coli or Salmonella.
· Provide some comments about how the pH-dependent torque-speed relationship and stator function of H. pylori flagellar motors might be impacted based on your rotation rate data.
· Indicate more clearly in Figure 7 which bacteria are stuck vs swimming and highlight trajectories.
Reviewer 3 Report
Comments and Suggestions for Authors
The manuscript entitled. Unraveling the intertwined effect of pH on Helicobacter pylori motility and the microrheology of the mucin-based medium it swims in. In this study phase contrast microscopy was used to track the movement of the bacteria and see the effects of acidic pH on the viscoelastic properties of the mucin-based mucus and the flagellar motors and chemo-receptors of the H. pylori. The manuscript is well written and has merit for publication. Below are a few queries which authors should address:
1. During the culture of Helicobacter pylori why 5 – 12% CO2 condition was applied infact growth of Helicobacter pylori is promoted by atmospheric oxygen levels in the presence of 10% CO2
2. Figure 6A can authors replace this figure with high-resolution
3. Author must provide a conclusion of the study after the discussion.
The manuscript should be accepted after the minor revision.
